# An expert judgment model to predict early stages of the COVID-19 pandemic in the United States

**Thomas McAndrew** [1] *, **Nicholas G. Reich** [2]

**1** Department of Community and Population Health, College of Health, Lehigh University, Bethlehem, Pennsylvania, United States of America, **2** Department of Biostatistics and Epidemiology, University of Massachusetts Amherst School of Public Health and Health Sciences, Amherst, Massachusetts, United States of America

* mcandrew@lehigh.edu

**Data Availability Statement:** A publicly available repository with details about questions asked and data on all responses is available under a MIT license at https://github.com/tomcm39/COVID19_expert_survey.

## Abstract

From February to May 2020, experts in the modeling of infectious disease provided quantitative predictions and estimates of trends in the emerging COVID-19 pandemic in a series of 13 surveys. Data on existing transmission patterns were sparse when the pandemic began, but experts synthesized information available to them to provide quantitative, judgment-based assessments of the current and future state of the pandemic. We aggregated expert predictions into a single "linear pool" by taking an equally weighted average of their probabilistic statements. At a time when few computational models made public estimates or predictions about the pandemic, expert judgment provided (a) falsifiable predictions of short- and long-term pandemic outcomes related to reported COVID-19 cases, hospitalizations, and deaths, (b) estimates of latent viral transmission, and (c) counterfactual assessments of pandemic trajectories under different scenarios. The linear pool approach of aggregating expert predictions provided more consistently accurate predictions than any individual expert, although the predictive accuracy of a linear pool rarely provided the most accurate prediction. This work highlights the importance that an expert linear pool could play in flexibly assessing a wide array of risks early in future emerging outbreaks, especially in settings where available data cannot yet support data-driven computational modeling.

## Author summary

We asked experts in the modeling of infectious disease to submit probabilistic predictions of the spread and burden of SARS-CoV-2/COVID-19 from February to May, 2020 in an effort to support public health decision making. Expert predictions were aggregated into a linear pool. We found experts could produce short and long term predictions related to the pandemic that could be compared to ground truth such as the number of cases occurring by the end of the week and predictions of unmeasurable outcomes such as latent viral transmission. Experts were also able to make counter factual predictions—predictions of an outcome assuming an action will continue or not continue. In addition, predictions

**Funding:** This work has been supported by the National Institutes of General Medical Sciences (NIGMS, grant number R35GM119582) [NGR] and the Centers for Disease Control and Prevention (CDC, grant number 1U01IP001122) [NGR]. The content is solely the responsibility of the authors and does not necessarily represent the official views of CDC, NIGMS, or the National Institutes of Health. The funders had no role in study design, data collection and analysis, decision to publish, or preparation of the manuscript.

**Competing interests:** The authors have declared that no competing interests exist.

built by aggregating individual expert predictions were less variable when compared to predictions made by individuals. Our work highlights that an expert linear pool is a fast, flexible tool that can support situational awareness for public health officials during an emerging outbreak.

## Introduction

The first COVID-19 cases globally were reported in December of 2019 [1]. The World Health Organization (WHO) declared the outbreak a Public Health Emergency of International Concern on January 30, 2020, and on March 11, 2020, after the virus began spreading to other continents [2, 3], the WHO designated the outbreak a pandemic [4]. The first COVID-19 case in the United States without known origin occurred in Washington state in late January [1].

As with previous outbreaks of other diseases [5–7], forecasts from computational models [8–11] assisted in planning and outbreak response near the beginning of the pandemic. However, given the initial limitations in testing capacity for SARS-CoV-2, these models were confronted with imperfect data with which to explain and predict viral transmission dynamics. Because of these challenges in the early phase of the pandemic, some models faced criticism for a lack of accuracy [12].

Starting in mid-February 2020, shortly after the first US case of COVID-19 was identified and before any large-scale computational modeling efforts were public in the US, we collected and aggregated probabilistic predictions and estimates—quantitative statements provided as a probability distribution over potential future values—in weekly surveys of experts in the modeling of infectious disease [13].

Human predictions have been able to accurately predict phenomena or support the development of a predictive model in many domains from ecology to economics [14–21]. Expert and non-expert crowds have made accurate predictions of sleep disturbances [16], geopolitical events [15, 17], and meteorologic events [20, 21], and experts have successfully chosen variables that improved the prediction of clinical phenomena [18, 22]. In the context of infectious disease, human judgment (from a knowledgeable but not exclusively "expert" panel) produced accurate forecasts of seasonal influenza outbreaks in recent seasons [5].

However, there is as much past work highlighting experts' predictive skill as there is highlighting their difficulty in accurately assessing uncertainty [23–25]. Differences in predictive performance are thought to be a result of individual experts' abilities and how they interact with their environment–the probabilistic relationships between cues (data and signals available to them to help form an informed judgment) and the target of interest [26–33]. Past work has shown expert performance decreases when tasks are complex, ambiguous, and when experts receive imprecise feedback [34–37]. Expert performance also decreases when there is an overabundance of information in the environment potentially related to the target of interest [36]. Experts are also not exempt from cognitive biases that impact human judgment [23, 38–41]. Gains in predictive accuracy have been found when individual human predictions are combined into a linear pool [42–44].

Recent work in human judgement and crowdsourcing applied techniques from expert elicitation and decision theory to computational models of COVID-19 [45], crowdsourcing antiviral drugs to inhibit SARS-CoV-2 [46], and have used social media to collect patient-level data on COVID-19 to track transmission [47]. Predictions of COVID-19 from experts and lay persons have been compared, finding experts can make more accurate, though often overconfident, predictions [48]. Particular to COVID-19, human judgment and crowdsourcing are

methods for rapidly collecting and organizing data, and exploring many potential interventions in parallel.

In the early months of the pandemic there was a tremendous amount of information being generated. Between pre-print scientific research, media attention and amplification of both accurate and inaccurate findings, sparse past examples of an outbreak of this magnitude, and fast-changing government responses, assessing the current state and predicting the future trajectory of the pandemic were very difficult tasks in early 2020. This work assesses the performance of a panel of experts in providing quantitative estimates of a wide range of at-the-time unknown quantities relating to the scale and pace of the emerging COVID-19 pandemic in the US.

## Materials and methods

### Ethics statement

The proposed research was deemed not human subjects research by the University of Massachusetts-Amherst Institutional Review Board (IRB Determination Number 20–54).

### Recruitment of experts

We defined an expert as a researcher who has spent a substantial amount of time in their professional career designing, building, and/or interpreting models to explain and understand infectious disease dynamics and/or the associated policy implications in human populations.

Experts were recruited by sending an email asking for their participation, and by soliciting participation through online forums for infectious disease modelers.

Experts could participate in our surveys after reading and agreeing to a consent document (S1 Fig). The consent states that after an expert completed two surveys their name and affiliation would be included in public-facing summaries. The consent form also said that public releases of the data would ensure individual expert responses would remain unidentifiable. A list of experts who participated in two or more surveys can be found in the data repository [13] and in S3 Table in the supplement.

Survey data were collected using the web-based Qualtrics platform (Qualtrics, Seattle, WA) through a link sent via email. A link to the survey was also placed on an online forum of modelers focused on COVID-19, asking a participant to self-identify as an expert and fill out the survey. If the participant was vetted to be an expert by the research team, according to the expert definition above, they were added to the list of those that receive weekly emails. Predictions were collected from experts starting on the Monday of each week and closing on Tuesday of that same week.

### Survey methods

We asked experts questions that required them to submit predictions in one of four different formats: (i) categorical questions asked experts to pick one out of two (binary) or many (categorical) options, (ii) probabilistic questions asked experts to assign a probability to two (binary probabilistic) or many (categorical probabilistic) options, (iii) percentile questions asked experts to provide a lower (5th or 10th) percentile, median percentile, and upper (90th or 95th) percentile, (iv) triplet questions asked experts to report a smallest, most likely, and largest possible value for a forecasting target of interest. The format for how we asked experts to submit answers changed from smallest, most likely, and largest to a percentile format to more clearly communicate the predictions we meant to elicit from experts. A list of all questions and

the type of answer required by survey can be found in the data repository and in the supplement in S6 Table.

Data on true outcomes predicted by experts were collected from several sources (S1 Table). For questions that have known, measurable answers, we created a database with the observed answers and the resolution criteria—the method used to define the true answer—that can be found in the data repository.

**Survey of cases.**   For surveys administered from February 17 to April 6 we asked participants to provide the smallest, most likely, and largest possible number of cases that would occur by the end of the week, and for surveys administered from April 13 to May 11 participants were asked to assign probabilities to intervals where the number of cases could occur.

**Survey of deaths.**   For surveys administered on March 16, 23, 30, we asked participants to provide the smallest, most likely, and largest possible number of deaths that would occur by the end of 2020, on April 20th we asked participants to provide a 5th, 50th, and 95th percentile, and on May 4 we asked participants to provide a 10th, 50th, and 90th percentile.

**Surveys of latent viral transmission.**   For surveys administered from March 2nd to April 6 we asked experts to provide a smallest, most likely, and largest estimate. The last survey, on April 27, asked experts to provide a 10th, 50th, and 90th percentile. Experts were asked from March 2 to March 16 to predict the percent of confirmed cases, and from March 23 to April 27 experts were asked to predict the total number of infections.

**Surveys of counterfactual predictions.**   For a survey administered on April 27 we asked experts to provide a smallest, most likely, and largest estimate of the 7-day moving average of reported COVID-19 cases in Georgia (i) if the state continued to loosen restrictions and (ii) if the state did not loosen restrictions. A survey administered on May 4 asked experts to provide a smallest, most likely, and largest estimate of the 7-day moving average of reported COVID-19 cases in Texas if the state loosed or did not loosen restrictions. For a survey on May 11 we asked experts to provide a 10th, 50th and 90th percentile estimate of the 7-day moving average of reported COVID-19 cases in Washington state is the state began, or did not begin, an accelerated restart.

## Data repository

A publicly available repository with details about questions asked and data on all responses is available under a MIT license at https://github.com/tomcm39/COVID19_expert_survey.

## Statistical Methods

**Rounding.**   When reported in the text, expert predictions in the manuscript were taken to have two significant digits. For example, the number 12 was rounded to 12, 123 was rounded to 120, 1,234 was rounded to 1,200, 12,345 was rounded to 12,000, etc. We felt that rounding like this maintained the relevant precision in estimates for public health practice. Rounding was not performed when calculating scores.

**Linear pooling.**   A linear pool ($f$) is a function that takes as input $E$ expert predictions formatted as probability distributions $f_1, f_2, \cdots, f_E$ over a single target value $t$ [49]. We assume the true value $t$ was generated from a random variable $T$ whose distribution we do not know. The linear pool then outputs a probability distribution over $t$,

$$f(x) = \sum_{e=1}^{E} \pi_e f_e(x) \text{ such that } \sum_{e=1}^{E} \pi_e = 1$$

where $\pi_e$ is a weight between zero and one assigned to expert $e$. Because each $f_e$ is a probability

distribution the linear pool $f$ is a probability distribution. For this work we chose to weight all experts equally, assigning a weight of $\pi_e = 1/E$ to each expert (see S1 Appendix for additional weighting schemes and results).

**Scoring predictions.**   Expert predictions were scored against true outcomes using the log score [50, 51]. The log score is a proper scoring rule [50–52] that assigns the log of the probability a forecast placed on the true outcome

$$\mathrm{LS}(x) = \ln\left[p(x)\right]$$

where $p(x)$ is the forecasted probability assigned to the true value $x$. A log score of 0 indicates the forecast placed a probability of 1 on the truth and is the best score. A log score of negative infinity indicates the forecast placed a probability of 0 on the truth and is the worst score. Experts' scores for measurable questions are stored in the data repository [13].

A common transformation of the log score is forecast skill [53], defined as the exponentiated log score.

$$\mathrm{Forecast\ skill}(x) = \exp\{\mathrm{LS}(x)\} = p(x)$$

Linear pool performance was reported using the forecast skill of the linear pool divided by the forecast skill of an unskilled forecaster minus one, or the relative difference in forecast skill compared to an unskilled forecaster.

$$\mathrm{Relative\ skill} = \frac{\mathrm{Forecast\ skill\ of\ model}}{\mathrm{Forecast\ skill\ of\ unskilled\ forecaster}} - 1$$

A positive difference indicates the linear pool or individual expert is better informed than an unskilled forecaster.

Given $N$ predictions, forecast skill percentile is computed for each prediction by ranking all $N$ predictions by their forecast skill, assigning a value of one to the smallest forecast skill and a value of $N$ to the largest forecast skill, dividing by the number of predictions ($N$), and multiplying by 100. A forecast skill percentile of 0 is the least accurate prediction and a forecast skill percentile of 100 is the most accurate prediction.

We used the concept of an "unskilled forecaster" as a benchmark against which to measure predictive performance. For triplet questions, an unskilled forecaster is one who assigns a uniform probability mass to all values between the lowest and the highest predictions any expert proposed for a particular question. An unskilled percentile question assigned to its lower percentile the minimum of all lower percentiles, and to its upper percentile the maximum of all largest percentiles. The unskilled median was the median of all median percentiles. For probabilistic forecasts, an unskilled forecaster assigns $1/C$, where $C$ is the number of categories, to each category available as an answer.

**From expert predictions to probabilistic forecasts.**   Experts made four types of probabilistic predictions: binary, categorical, percentile, and triplet. For binary probabilistic questions, we can define the expert's prediction of the presence of an event as $p$. Then an expert's predictive distribution is Bernoulli($p$). A similar approach can be taken for categorical probabilistic questions, and we define an expert's predictive distribution over $C$ different choices as a Multinomial($N{=}1, p_1, p_2, \cdots, p_C$) distribution. Percentile questions ask experts to provide three percentiles, a low (5th or 10th) percentile, 50th percentile (median), and high (90th or 95th) percentile. The lowest (5th or 10th), middle (50th) and highest (90th or 95th) percentiles are called $p_{\mathrm{low}}, p_{\mathrm{middle}}, p_{\mathrm{high}}$, and the corresponding percentile values are called $q_{\mathrm{low}}, q_{\mathrm{middle}}, q_{\mathrm{high}}$. Specifying three percentiles creates four intervals with probabilities corresponding to the percentiles, for example asking for a 5th, 50th, and 95th percentile creates four intervals with the

following probabilities: 0.05,0.45,0.45, and 0.05 probability, and the probability prescribed to an integer value $x$ was

$$\text{PCT}(x|p_{\text{low}}, p_{\text{middle}}, p_{\text{high}}, q_{\text{low}}, q_{\text{middle}}, q_{\text{high}}) = \begin{cases} \dfrac{p_{\text{low}}}{q_{\text{low}}} & 0 \leq x < q_{\text{low}} \\[2ex] \dfrac{p_{\text{middle}}}{q_{\text{middle}} - q_{\text{low}}} & q_{\text{low}} \leq x < q_{\text{middle}} \\[2ex] \dfrac{p_{\text{high}}}{q_{\text{high}} - q_{\text{middle}}} & q_{\text{middle}} \leq x < q_{\text{high}} \\[2ex] \dfrac{p_{\text{high}}}{q_{\text{high}}} & q_{\text{high}} \leq x < 2q_{\text{high}} \\[2ex] 0 & \text{otherwise} \end{cases} \quad (1)$$

Triangular probability densities [54] (See S2 Fig for an example) were generated from the smallest ($s$), most likely ($m$), and largest ($l$) answer provided by experts as follows

$$\text{TPD}(x|s, m, l) = \begin{cases} 0 & x < s \\[2ex] \dfrac{2(x - s)}{(l - s)(m - s)} & s \leq x < m \\[2ex] \dfrac{2}{l - s} & x = m \\[2ex] \dfrac{2(l - x)}{(l - s)(l - m)} & m \leq x < l \\[2ex] 0 & x > l \end{cases} \quad (2)$$

The above TPD specifies an expert-specific probability distribution over a continuous target. To specify a distribution over integers values $(x_1, x_2, \cdots, x_V)$, we assigned to the value $x_i$ the integral of the continuous TPD from $x_i$ up to $x_{i+1}$. Define the CDF of a TPD distribution as $\text{CDF}_{\text{TPD}}(x)$. Then the discretized TPD was defined as

$$p(x_i) = \text{CDF}_{\text{TPD}}(x_{i+1}) - \text{CDF}_{\text{TPD}}(x)$$

for $i < V$ on the values $(x_1, x_2, \cdots, x_V)$.

**Imputing quantiles from a binned distribution.** Given a set of $N$ bins $[x_0, x_1], [x_0, x_1], \cdots, [x_{N-1}, x_N]$ with corresponding probabilities $p_1, p_2, \cdots, p_N$ over a target that takes nonnegative values, quantiles can be imputed by first generating a function f that is a linear interpolant of (x,y) pairs $(x_0, 0), (x_1, P_1), (x_2, P_2), (x_3, P_3), \cdots, (x_N, 1)$, where $i$ ranges from 0 to $N$ and $P_i$ is the cumulative sum of all probabilities with index $i$ or smaller $P_i = p_1 + p_2 + \cdots + p_i$. To compute a quantile $q$ we can find the root of the function $F(x) = f(x) - q$ using the Newton-Raphson algorithm (or any standard root finding procedure).

**Relative absolute error.** The relative absolute error is a function that takes as input a point prediction ($p$) and ground truth value ($t$) and outputs the absolute difference between the point prediction and ground truth value divided by the ground truth value

$$\text{Rel.Abs.Err}(p, t) = \left| \frac{p}{t} - 1 \right|$$

where the vertical bars indicate the absolute value. The domain of this function is defined only for positive values of $t$ (i.e. $t > 0$).

## Results

### Overview

Thirteen surveys of experts in the modeling of infectious disease were conducted between February 18, 2020 and May 11, 2020 (see S1 Table for schedule, S3 Table for a list of experts who participated, and S6 Table for a list of all questions) [13]. We solicited the participation of 72 experts (see Methods for definition) via email and a Slack channel dedicated to communicating about COVID-19 data and models. A total of 41 experts contributed predictions, with an average of 18.6 expert participants each week (range: 15–22).

Across all surveys, we asked 73 questions (48 with measurable outcomes) with a median of 6 questions per survey (range: 4–7) focused on the outbreak in the United States. Experts responded to questions on a variety of topics including short- and long-term predictions of COVID-19 cases, hospitalizations, and deaths, and we combined predictions into linear pools. The survey results were released publicly every week and delivered directly to decision makers at state and federal health agencies (see [13] for copies of each summary report).

Here we present results from selected questions. We chose to report on questions that other computational models have tried to predict: the number of deaths due to COVID-19 in the US by the end of 2020, the total number of SARS-COV-2 infections in the US, and the number of confirmed cases one week ahead. Additionally, we include some results from counterfactual questions that asked experts to predict the 7-day rolling average number of cases for a state under the current policies in place and if the state increased restrictions on social contact. Expert linear pool estimates and predictions for all questions are stored in a public GitHub repository [13].

### Predictions of US COVID-19 deaths reported in 2020

Across five surveys administered between March 16 and May 4, we asked experts to predict the number of reported COVID-19 deaths in the US by the end of 2020. The linear pool prediction ranged from 150,000 to more than 250,000 (Fig 1A), corresponding to between 4 and 7 times the average number of annual deaths in the US due to seasonal influenza [55]. There was considerable uncertainty around these predictions: the lower bounds of the five prediction intervals ranged from 6,000 (on March 16) to 118,000 (on May 4), and the upper bounds ranged from 517,000 (on April 20) to 1,700,000 (on March 30). The COVID Tracking Project reported that there were 336,802 cumulative deaths due to COVID-19 in the US at the end of Dec. 31 2020. This eventually observed value was included in the 90% prediction interval for each of the five surveys. While experts underestimated the actual number of observed deaths by a substantial margin, they also consistently saw this eventual number of deaths as a not unlikely outcome, assigning a probability of 0.45 on March 16 and 0.46 on March 30 to more than 250K deaths.

### Predictions of weekly COVID-19 reported cases

At the beginning of thirteen consecutive weeks from February 17 to May 11, experts predicted the number of confirmed cases at the end of the week (Fig 1B). In early surveys, experts tended to underestimate the number of reported cases in the following week. The relative difference between the median linear pool prediction of week-ahead cases and the reported cases was on average -51% for the first four surveys. In later surveys, the accuracy improved and for surveys

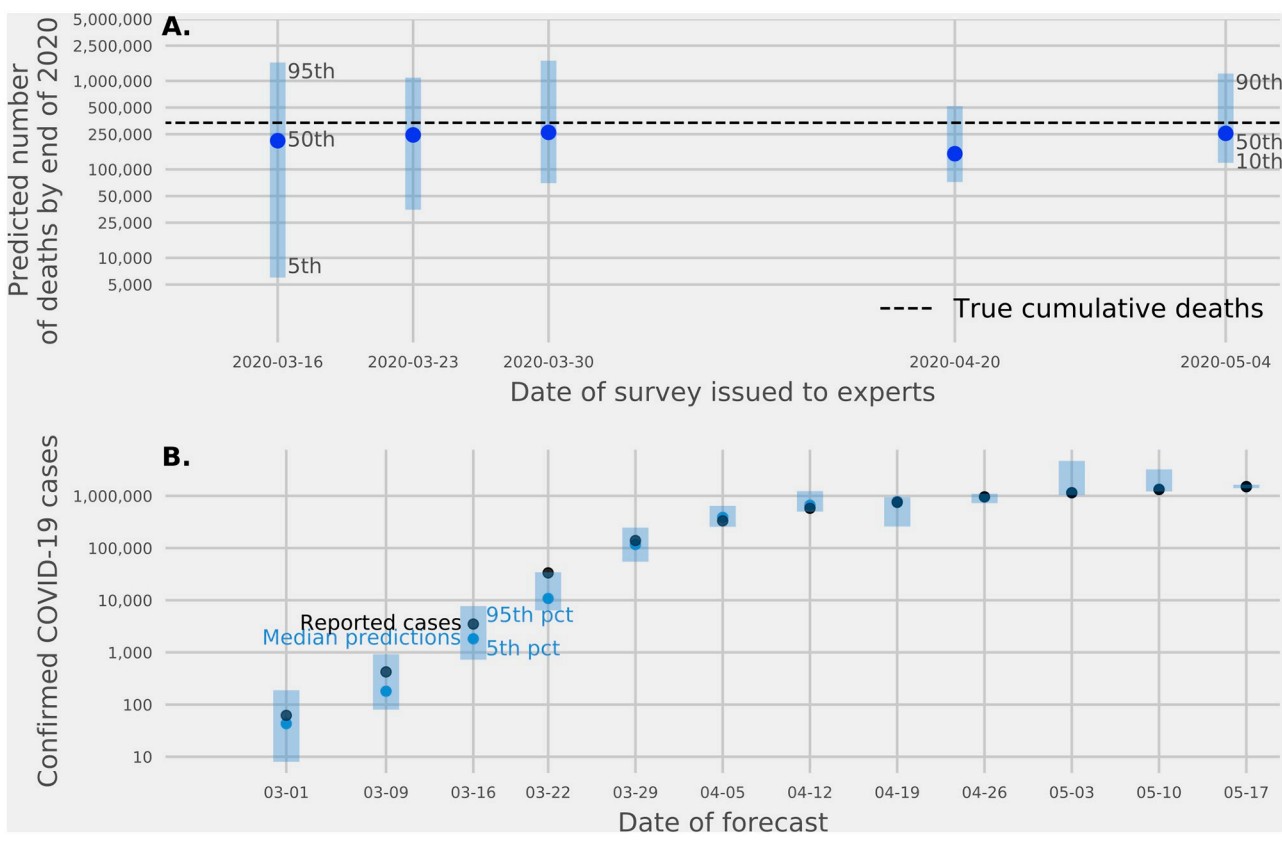

**Fig 1. Expert predictions of confirmed COVID-19 cases and deaths.** (A.) Expert linear pool predictions of the total number of deaths by the end of 2020 from five surveys asked between March 16 and May 4, 2020. Points show the median estimate. Bars show 90% prediction intervals for the first four surveys and an 80% prediction interval for the fifth survey. The dotted line is the reported total number of deaths by The COVID Tracking Project as of December 31, 2020. (B.) Expert linear pool forecasts, made on Monday and Tuesdays, of the number of cases to be reported by the end of the week (Sunday, date shown on x-axis) from thirteen surveys administered between February 23 and May 17, 2020. The first eight surveys asked experts to provide smallest, most likely, and largest possible values for the number of confirmed cases, and the last five asked experts to assign probabilities to ranges of values for confirmed cases. Light blue points represent the median of the expert linear pool distribution. Dark points represent the eventually observed value reported by The COVID Tracking Project. Prediction intervals at the 90% level are shown in shaded blue bars. The 90% prediction intervals included the true number of cases in all thirteen forecasts.

5 through 13, the relative difference was on average 2.5%. However, as with the forecasts of total deaths, the expert linear pool provided wide uncertainty: all thirteen expert linear pool 90% confidence intervals covered the reported number of confirmed cases.

## Estimates of the fraction of infections reported as cases

Over the course of seven surveys from March 2 to April 27, 2020, experts were asked to estimate the fraction of all infections with SARS-CoV-2 (the virus that causes the COVID-19 illness) in the U.S. that had been confirmed and reported as a case. Because this outcome can never be fully observed, these questions posed a different type of estimation task for the experts.

In these surveys, the median of expert linear pool predictions for the percentage of infections detected was between 6% and 16% (Fig 2B). The median responses were consistent with estimates from contemporaneous computational models, perhaps reflecting the extent to which experts relied on these early model estimates [9–11, 56] (Fig 2A).

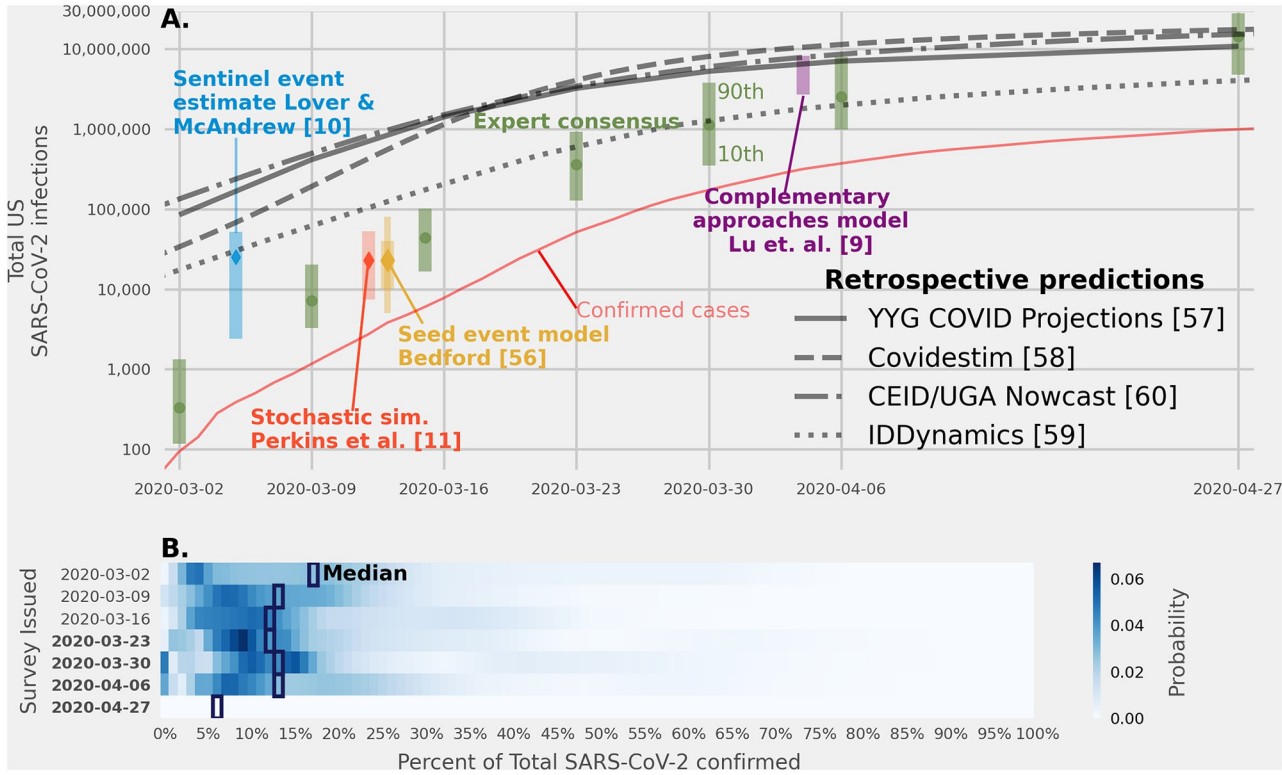

**Fig 2. Expert predictions of total number of SARS-CoV-2 infections.** Estimates of the total number of infections with the SARS-CoV-2 virus, made in different weeks across early 2020. (A.) Estimates of the total number of SARS-CoV-2 infections (both observed and unobserved) from the expert linear pool model (green dots, with 80% prediction intervals), four contemporaneous estimates (blue, red, yellow, and purple bars), and four retrospective estimates from computational models (fit in 2021) [9–11, 57–60]. Prediction intervals at the 80% level are shown for all prospective estimates except for the Bedford estimate which provided a "best guess" prediction interval and a second interval double the size of the first shown as a narrower line [56]. Expert predictions aligned with contemporaneous model estimates throughout the entire survey time. Real-time estimates from both models and expert linear pools were 2–3 orders of magnitude smaller than retrospective model estimates (estimates generated in early 2021) for early March 2020 and more in alignment by the end of April 2020. (B.) Expert linear pool distributions of the fraction of all infections reported as confirmed cases. In the first three surveys, experts provided a predicted percent of infections that had been confirmed as cases by laboratory test, in the next four (dates in boldface) they directly estimated the total number of infections. Surveys 4–6 asked experts to provide the smallest, most likely, and highest number of total infections, and the last survey asked experts to provide a 10th, 50th, and 90th percentile (shadings not included in the figure).

However, these estimates from computational models and expert judgment made in early 2020 were substantially smaller than 3 out of 4 retrospective estimates of the total number of SARS-CoV-2 infections that were made in early 2021 [57–60]. Some of these model-based estimates suggested that the detected fraction of cases was possibly as low as 0.1%, although substantial uncertainty was present even in retrospect [57–60].

## Counterfactual predictions

Experts were asked to make counterfactual predictions of the 7-day moving average of confirmed cases for three states that had begun to relax social distancing restrictions ("re-open"). The questions asked experts to predict how many confirmed cases each state would see between 3 and 5 weeks' time for each of two separate scenarios: (i) if the state continued its current phase of reopening and (ii) if the state did not begin to reopen (Fig 3).

Expert predictions showed a clear expectation that state-level policies restricting non-essential travel and business would result in lower COVID-19 transmission in coming weeks. The

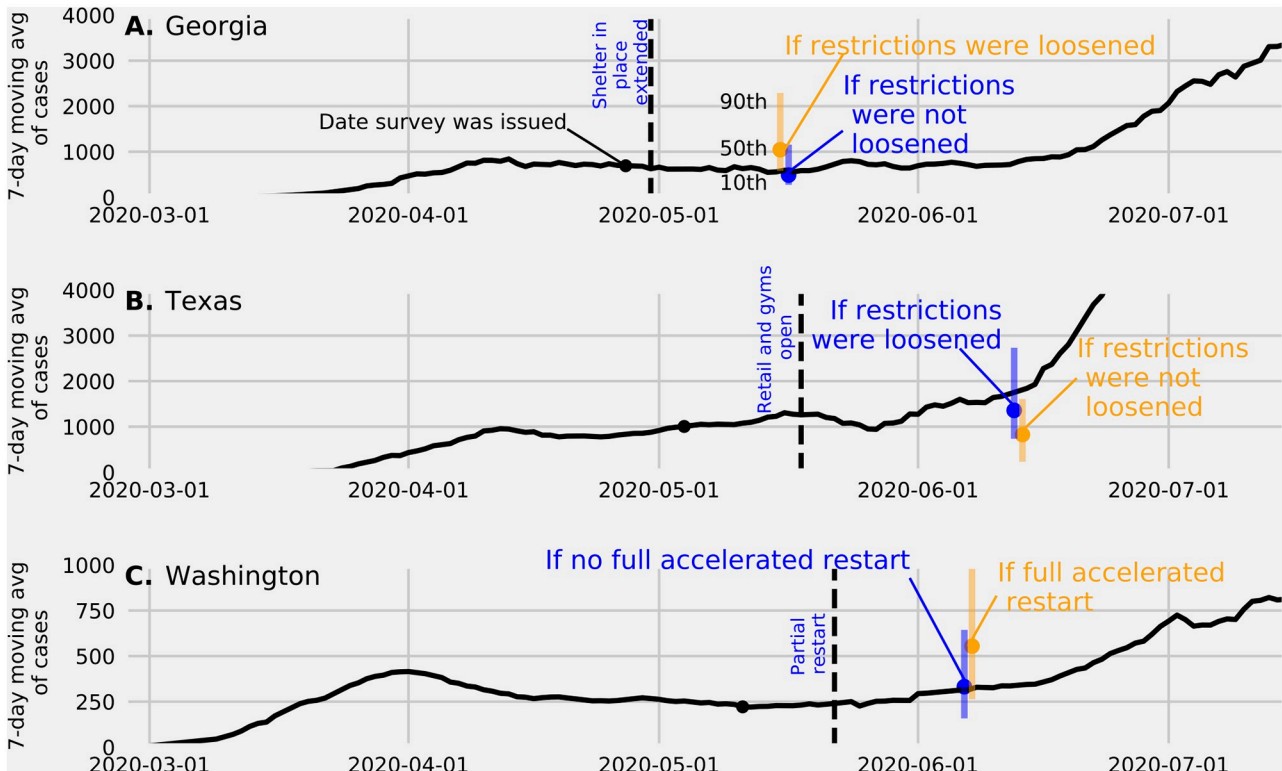

**Fig 3. Counterfactual predictions of reported COVID-19 cases.** Summaries of counterfactual predictions of reported COVID-19 cases made by experts for three states. In each panel, experts made predictions under an "optimistic" and "pessimistic" scenario about the impact of re-opening on COVID-19 transmission. All predictions were made about an outcome between 3–5 weeks into the future. The time at which predictions were made are shown with a black dot. The scenario that ended up being more aligned with reality at the target prediction date are shown in blue. The date at which relevant policies were enacted between the survey date and the resolution date are indicated with a vertical dashed line. (A). An expert linear pool median and 80% CI (10th percentile and 90th percentile) made on April 27, 2020 (black circle) of the 7- day moving average of reported COVID-19 cases for the state of Georgia for the week of May 10 to May 16, 2020 under two scenarios: if the state of Georgia reopens several businesses or "loosens restrictions" (orange) or if restriction were not loosened (blue). The 7-day moving average reported by the Georgia Department of Health is in black. (B). Expert linear pool predictions made on May 4, 2020 of the 7-day moving average of reported COVID-19 cases for the state of Texas for the week ending on June 13, 2020. Predictions were made under the differing assumptions that (i) Texas continued to loosen restrictions (blue) and (ii) that the state did not loosen restrictions (orange). The moving average reported by the Texas Department of Health is shown as the solid black line. Quantiles for the expert linear pool predictions were imputed by assuming a uniform distribution over values within 5 intervals. (C) Expert linear pool predictions made on May 11, 2020 of the 7-day moving average of reported COVID-19 cases for the state of Washington for the week ending on June 7, 2020. Predictions were made under two assumptions: (i) Washington would begin their Phase II plan on May 16, 2020, an accelerated restart, for all counties (orange) or Phase II would not begin by May 16 for all counties (blue). The moving average reported by the Washington Department of Health is shown as the solid black line.

expert linear pool predictions for the scenarios that were more clearly aligned with the eventual reopening policies were more accurate than for the alternative scenarios.

For the state of Georgia (Fig 3A) the expert linear pool prediction on April 27, 2020 of the 7-day moving average of reported COVID-19 cases on May 16, 2020 was 1,044 (80% PI = [580, 2, 292]) assuming restrictions were loosened and 487 (80% PI = [273, 1, 156]) assuming restrictions were not loosened. On Apr 30, 2020 the state decided to extend shelter in place orders for at risk populations. The relative absolute error of the linear pool prediction was 18% for that scenario versus 75% for the scenario assuming restrictions were loosened. For the state of Texas (Fig 3B), the expert linear pool prediction on May 4, 2020 of the 7-day moving average of reported COVID-19 cases on June 13, 2020 was 825 (80% PI = [231, 1, 608]) assuming restrictions were not loosened and 1,358 (80% PI = [734, 2, 732]) assuming restrictions were

loosened. On May 18, 2020 the state decided to reopen retail businesses. The relative absolute error of the linear pool prediction was 23% for that scenario versus 54% for the scenario assuming restrictions were not loosened. For the state of Washington (Fig 3C) the expert linear pool median prediction on May 11, 2020 of the 7-day moving average of reported COVID-19 cases was 332 (80% PI = [158, 644]) if WA did not begin an "accelerated restart" by relaxing restrictions in all counties and 554 (80% PI = [263, 1, 053]) if it did. On May 22, 2020 the state decided to reopen 25/39 counties. The relative absolute error of the linear pool prediction was 5% for a partial reopening versus 76% for the scenario assuming an accelerated restart in all counties.

## Individual experts' predictive performance varied substantially

The median forecast skill of individual expert predictions of the number of deaths by Dec 31, 2020 was above that of an unskilled forecaster for all five surveys where this question was asked (Fig 4A). The median forecast skill of individual expert predictions of new cases in the

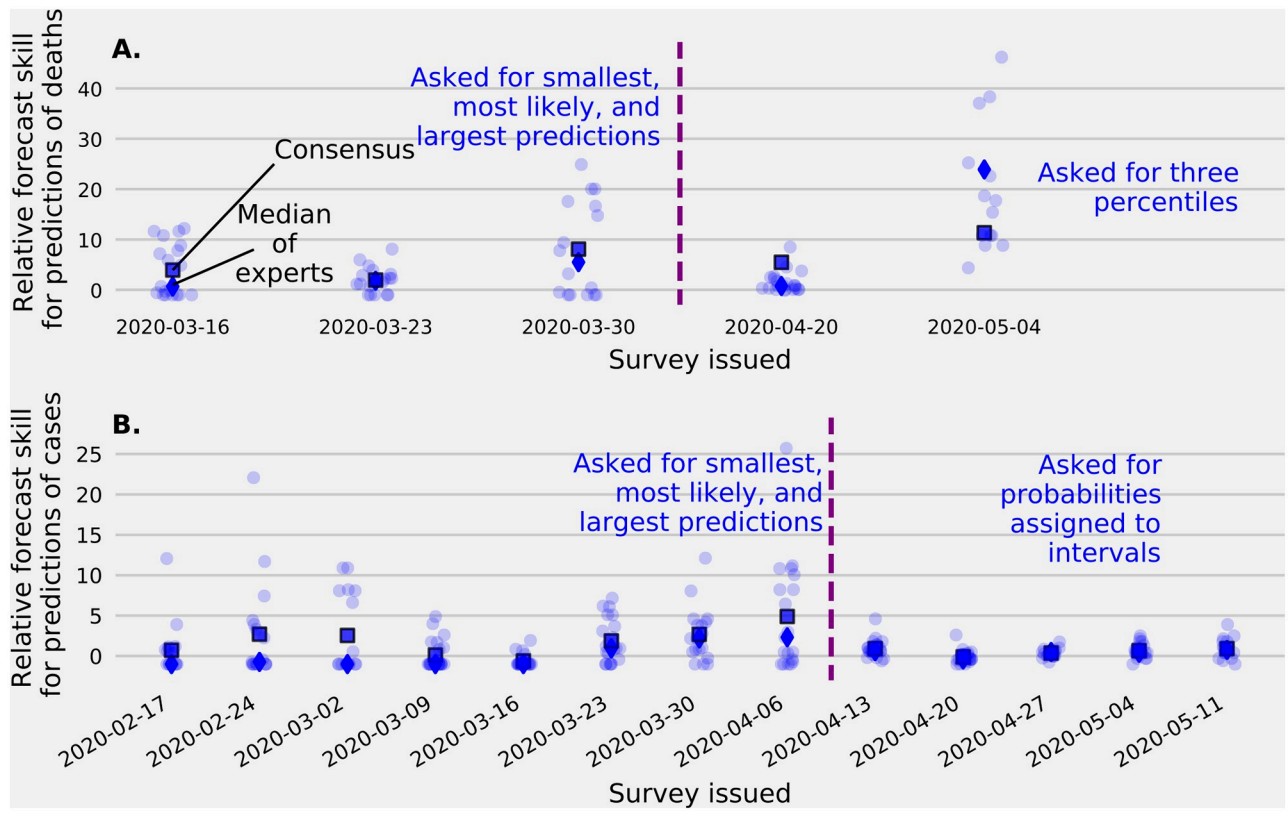

**Fig 4. Forecast accuracy for expert predictions of cases and deaths.** Evaluation of forecast accuracy for forecasts of cumulative COVID-19 deaths (A) and cases (B). For both types of questions, the methods used to elicit probabilistic forecasts changed and this point is indicated by a vertical dashed line. Predictions are shown from each expert (light dots), the median expert (dark diamond), and the linear pool (dark square) compared to an "unskilled" forecaster (see Methods). Higher relative forecast skill indicates better performance than an unskilled forecaster and a zero relative forecast skill represents identical performance with an unskilled forecaster. (A). Relative forecast skill of the cumulative number of COVID-19 deaths by December 31, 2020 (see Fig 1A). Over 50% of experts made better predictions of year-end COVID-19 deaths than an unskilled forecaster on each of the five occasions this question was asked. Experts' median relative forecast skill was higher than the linear pool forecast skill for the latest prediction of year-end deaths when asked to provide percentiles compared to the smallest, most likely, and largest number of deaths. (B.) Relative forecast skill of the number of cases to be reported by the end of the week from thirteen surveys administered between February 23 and May 17, 2020. Individual experts' accuracy was mixed with some experts performing better than an unskilled forecaster and others scoring worse. In the first five surveys, the median expert made less skilled forecasts than the unskilled forecaster. Experts' median relative forecast skill was smaller than the linear pool forecast skill when asked to provide a smallest, most likely, and largest number of cases and similar to a linear pool when asked to assign probabilities to a set of intervals where the true number of cases could fall.

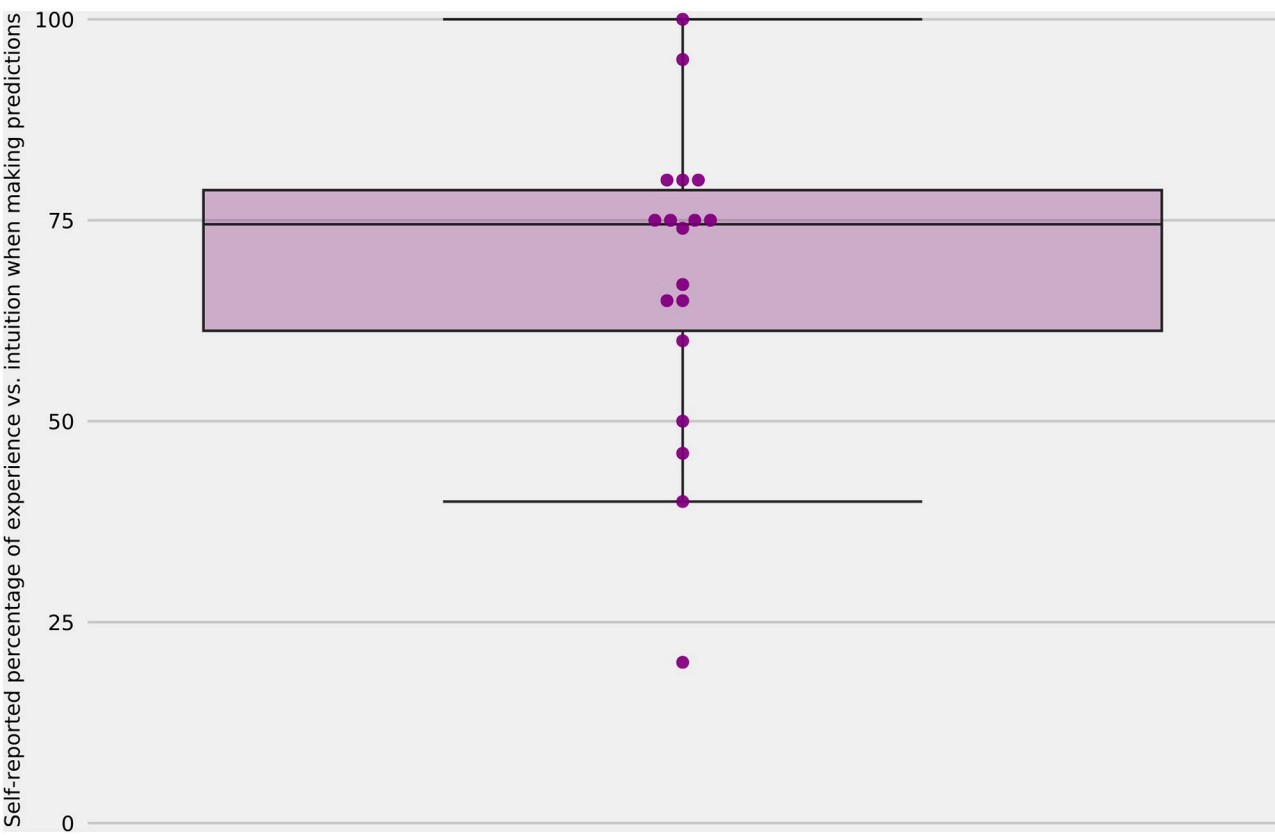

**Fig 5. Expert analytic versus intuitive thinking.** Experts' self-assessed percentage of analytic vs. intuitive thinking when making predictions, reporting 0 when an expert uses only their intuition and 100 when they relied solely on models and experience. To make predictions over a wide variety of targets, experts reported a mixture of using models/experience and intuition, with the median expert claiming to rely 75% on experience.

coming week was lower than an unskilled forecaster on the first five surveys (Fig 4B). Forecasts of cases from individual experts were more accurate in later surveys, and the median accuracy of individual expert predictions was higher than the accuracy of an unskilled forecaster in 7 of the last 8 surveys.

Because expert performance was not consistent across surveys, performance-based weighting to build a linear pool did not significantly improve forecast accuracy compared to equal weighting (see supplemental section on aggregation, S4 Fig and S5 Fig, and S4 Table and S5 Table). The degree to which experts relied on available data, model outputs, and intuition varied by expert. The median proportion of each prediction that relied on analytic models versus intuition which was self-reported by experts was 75% and responses ranged from 20% to 100% (Fig 5).

### Accuracy of the expert linear pool

Looking across all questions with measurable probabilistic outcomes from February 17 to May 11, the linear pool prediction was the most consistently accurate forecast. When ranked alongside all individual expert predictions, the linear pool was among the top 50% most accurate forecasts 36/44 (82%) times (Fig 6A). The linear pool mean forecast skill percentile of 73 was the highest when compared to the mean forecast skill percentile of experts who completed ten or more surveys (Fig 6B). Over all 13 surveys issued, the linear pool model ranked in the top

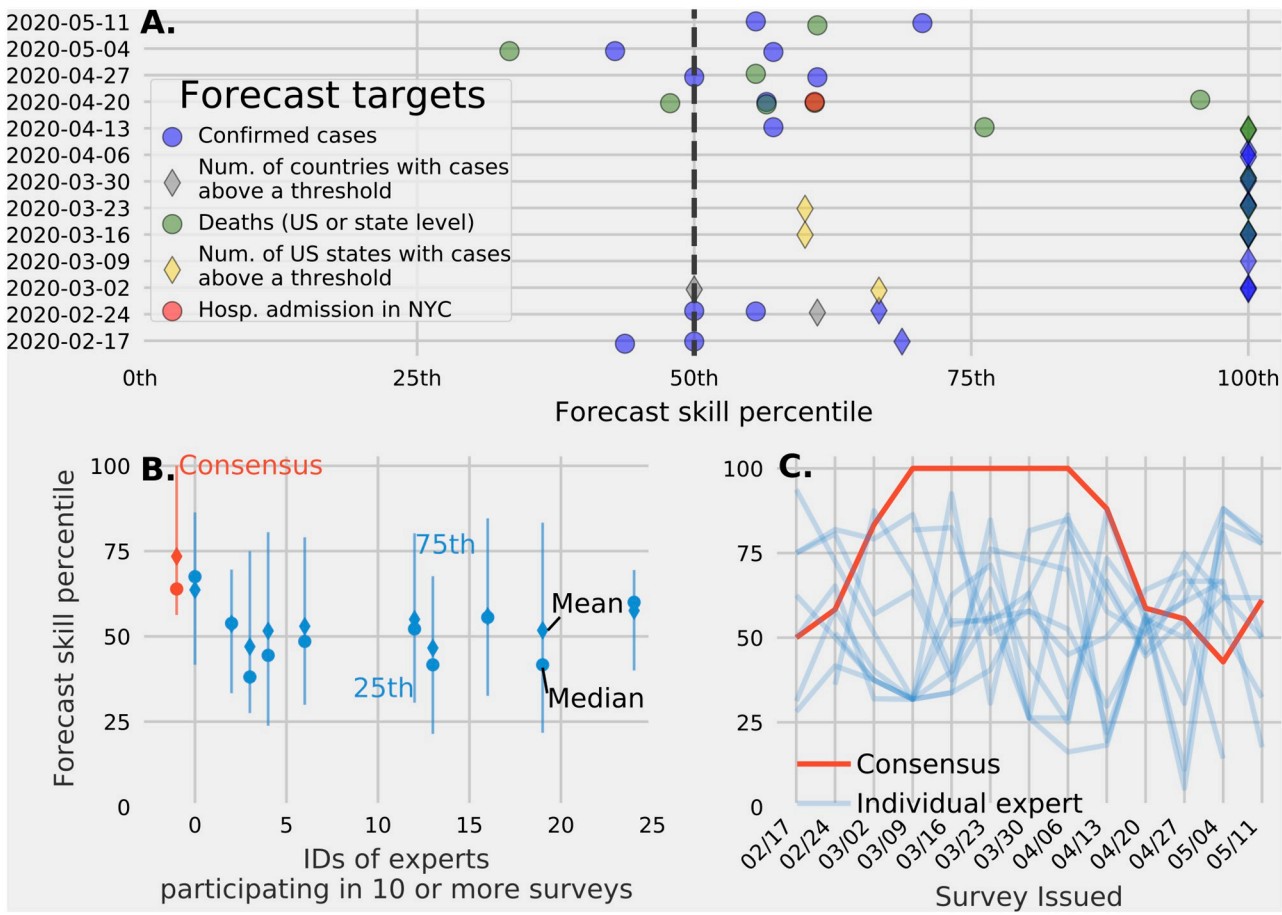

**Fig 6. Expert linear pool forecast skill.** The forecast skill percentile of linear pool predictions compared to individual expert predictions across all 13 surveys (vertical axis) differentiated by the type of target. A linear pool of expert judgment often scores in the top 50th percentile independent of the type of question (B.) The mean, median, 25th, and 75th percentile for forecast skill percentile for all individual experts who completed 10 or more surveys (blue) and for the linear pool (red). Compared to individual experts, a linear pool has the highest mean and highest 25th percentile forecast skill percentile. (C.) The median forecast skill percentile across surveys for experts (blue) and the linear pool (red). Over time the linear pool median forecast skill percentile is above 0.50 for all but one survey and for five surveys the linear pool generated the most accurate predictions.

half of all experts and was the most accurate for five surveys issued from May 9 to April 6 (Fig 6C). The linear pool performance depended on the structure of the questions, which changed over the course of the surveys (e.g., see Fig 4, Methods, and a list of all survey questions in the supplement).

## Discussion

Linear pool aggregations of expert judgment during early months of the COVID-19 pandemic provided important and early insights about the trajectory of the emerging pandemic. In mid-March, when there were less than 100 COVID-19 deaths in the US, the expert linear pool assigned a probability of 66% to over 100,000 deaths and a probability of 46% to over 250,000 deaths by the end of 2020. In contrast, early forecasts from a computational model used by the federal government in late March predicted 81,000 deaths and an outbreak that would end by early August, 2020 [61].

An expert linear pool is a nimble and flexible model that can answer questions about the public health impact of outbreaks before computational models have enough data and

validation to be reliable. In particular, an expert linear pool model has two key advantages over computational models. First, an expert model has relatively low overhead to develop and can be deployed at the onset of an outbreak. The first expert predictions from these surveys were available starting in mid-February, 2020 before any computational models were publicly available. Second, a survey framework allowed expert predictions to be tailored on-the-fly to maximize value for public health decision makers. This is in contrast to computational models which require extensive development to answer a specific set of questions.

However, an expert linear pool model suffers from issues of scalability. Every individual forecast elicited from an expert requires minutes of human time. Because of experts' limited time, surveys must focus on a short list of impactful questions. Another limitation of an expert model is the bias introduced by human judgment. Though the assumptions built into computational models are explicitly specified, experts' predictive processes are more opaque. A structured forecasting platform, that allows experts to communicate with one another about the reasoning behind their forecasts and facilitates interactions between subject matter experts and trained forecasters, has been shown to lead to more accurate predictions [17]. Cultivating and training a pool of expert forecasters should be included as part of larger investments in preparing modeling infrastructure for future outbreaks and pandemics.

Experts reported using a combination of analytic thinking and intuition to form their responses when making predictions about targets related to the pandemic. The mix of analytic thinking and intuition could be because of the lack of structured data early during the pandemic or could be because different questions induced different modes of cognition [62–64]. Insight into the type of thinking experts prioritize when making a prediction and how these modes of cognition lead to more or less accurate predictions may allow us to develop elicitation protocols to either emphasize, or discourage specific approaches to forming a prediction. Expert predictions suggest they were able to synthesize diverse and disparate sources of information to make quantitative predictions and estimates about different facets of the pandemic, including short- and long-term predictions of observable data, short-term projections of counterfactual scenarios, and estimates of quantities that will never be fully observed.

Expert predictions of the US COVID-19 outbreak often outperformed an unskilled forecaster but in absolute terms were typically biased towards optimistic responses. Expert predictions of confirmed cases included the true number of cases in their 90% prediction interval for all thirteen predictions, a sign that the uncertainty reflected in the linear pool was perhaps too broad. Experts' predictions in early surveys were smaller than the true number of confirmed cases, but after receiving weekly feedback on their previous predictions (starting on March 9), accuracy improved.

However, the extent to which strong conclusions about expert prediction accuracy can be drawn from these data is constrained due to several key limitations of the present study. First, the sample size of the study is small, with only 73 questions asked and 48 questions with ground truth available. To fully assess probabilistic accuracy of a system making repeated predictions, larger numbers of questions are needed. Second, the type of answer and the type of target experts were asked to provide varied across the surveys. This was the result of the study organizers adapting to new information and phases of the early outbreak, and trying out different strategies for answering questions. Results appear to suggest (see, e.g., Fig 4) that assessments of individual expert accuracy and the variability in individual scores may depend on the way in which probability distributions were elicited. There are several elicitation protocols that could have been used to extract less biased and more informative predictions from experts [65, 66]. Third, due to the operational challenges that accompanied standing up this project in real-time during the early months of 2020, the pool of experts may not represent a full spectrum of expert opinion from the modeling community.

Despite the limitations outlined above, the present study shows the potential for a more structured and larger-scale effort to use expert judgment to supplement output from computational models. An expert judgment model can act as an important component of rapid response and as a first-step forecast for global catastrophes like an outbreak, especially while domain-specific computational models are still being trained on sparse early data. Experts' ability to synthesize diverse sources of information gives them a unique, complementary perspective to model-driven forecasts that are not able to assimilate information or data outside of the domain of a specific, prescribed computational framework.

During the evolving global catastrophe of the COVID-19 pandemic, an expert judgment model provided rapid and calibrated forecasts that were responsive to changing public health needs. If and when modeling needs are assessed for future outbreaks, the successes and limitations of this project could be used to design future expert judgment panels.

## Supporting information

**S1 Fig. Consent form.** The consent form each expert was presented with and had to agree to before taking part in the survey. This document was shown for every survey.
(PDF)

**S2 Fig. Triangular probability distribution.** An example of transforming an expert's triplet answer (smallest: 10, most likely:30, largest: 50) to a probabilistic distribution.
(PDF)

**S3 Fig. Triangular probability distribution ensemble.** An example of 10 expert answers to a triplet question, their corresponding triangular probability distributions (TPDs), and an equally-weighted linear pool distribution (black) built from those TPDs.
(PDF)

**S4 Fig. An analysis of individual expert forecast skill.**
(PDF)

**S5 Fig. Ensemble accuracy and ensemble weights assigned to experts.**
(PDF)

**S1 Table. Survey meta information.** A listing of survey numbers, the date they were issued, information on expert participation, and the database(s) used to collect ground truth.
(PDF)

**S2 Table. Ensemble covariates.** Model and the covariates used to define the design matrix $X$ to weight experts.
(PDF)

**S3 Table. List of experts who participated in at least two surveys.**
(PDF)

**S4 Table. Linear regression that compares the weights assigned to each expert using the expert-specific performance weighting and assigning experts equal weights.**
(PDF)

**S5 Table. Linear regression that compares the weights assigned to each expert using the expert-specific plus relative entropy performance weighting and assigning experts equal weights.**
(PDF)

**S6 Table. Date each survey was conducted, the questions asked, and the format experts answered.**
(PDF)

**S1 Appendix. Methodology to aggregate expert probabilistic predictions.**
(PDF)

## Acknowledgments

We wish to thank all the experts who have participated, for offering their time and expertise to help us better understand the COVID-19 outbreak. We also thank Evan L. Ray for comments that improved this work.

## Author Contributions

**Conceptualization:** Thomas McAndrew, Nicholas G. Reich.

**Data curation:** Thomas McAndrew, Nicholas G. Reich.

**Formal analysis:** Thomas McAndrew, Nicholas G. Reich.

**Investigation:** Thomas McAndrew, Nicholas G. Reich.

**Methodology:** Thomas McAndrew, Nicholas G. Reich.

**Project administration:** Thomas McAndrew, Nicholas G. Reich.

**Resources:** Thomas McAndrew, Nicholas G. Reich.

**Software:** Thomas McAndrew, Nicholas G. Reich.

**Supervision:** Thomas McAndrew, Nicholas G. Reich.

**Validation:** Thomas McAndrew, Nicholas G. Reich.

**Visualization:** Thomas McAndrew, Nicholas G. Reich.

**Writing – original draft:** Thomas McAndrew, Nicholas G. Reich.

**Writing – review & editing:** Thomas McAndrew, Nicholas G. Reich.

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
