## [Decision Letter · Decision Letter 0]

10 Jan 2022

Dear Dr. McAndrew,

Thank you very much for submitting your manuscript "An expert judgment model to predict early stages of the COVID-19 pandemic in the United States" for consideration at PLOS Computational Biology. As with all papers reviewed by the journal, your manuscript was reviewed by members of the editorial board and by several independent reviewers. The reviewers appreciated the attention to an important topic. Based on the reviews, we are likely to accept this manuscript for publication, providing that you modify the manuscript according to the review recommendations.

In particular, we hope that the authors can consider Reviewer 1's questions about how the experts interpreted the questions and the implications this has for the results, as well as Reviewer 2's concerns about the quality of the literature review, the description of the methods in the supplement, and other requests for minor clarifications of figures and text. 

Sincerely,

Alison L. Hill

Associate Editor

PLOS Computational Biology

Tom Britton

Deputy Editor

PLOS Computational Biology

[LINK]

Reviewer's Responses to Questions

**Comments to the Authors:**

Reviewer #1: This is brilliant work and was a real pleasure to read. Thorough efforts evaluating how aggregated expert forecasts perform in real time, and how they can be aggregated in ways that facilitates more accurate predictions, is crucially important work, and no one has done it for COVID-19 more thoroughly than the authors. The extensive longitudinal nature of this work, and the authors' willingness to update their protocol as they continued when weaknesses were identified, makes it especially valuable. Excellent points made in discussion as well.

A few comments:

1) How experts interpreted the phrase "smallest, most likely, and largest possible number of cases" is hard to know: did they literally interpret that as the smallest/largest possible number of cases, such that they would be infinitely flabbergasted if the true number were below/above these extrema? or (probably more likely) did they actually tend to respond with the smallest they thought plausible, or even the smallest they thought likely? In the way that the triangular probability densities are constructed, it is assumed that the expert meant that there is literally 0 probability of the true value falling below their "smallest" or above their "largest" estimate. This is fine, but of course if the experts weren't being quite so literal in their interpretation of "smallest/largest possible", it will result in inadequate allocation of probability mass to outcomes beyond these bounds. It seems possible that the early underestimates might not have seemed quite as extreme if an elicitation protocol were used that took this into account. This should perhaps be briefly discussed. (But I don't want to let the experts off the hook too much for those early predictions, as it's clear that they were massive underestimates!)

2) Was experts’ self-assessed percentage of analytic vs. intuitive thinking predictive of performance? If so, mentioning this fact might help inform guidelines for what kinds of thinking experts might consider prioritizing when asked to make probabilistic estimates.

3) It is mentioned that experts spent their careers either working with models of infectious disease dynamics (I'll call these people 'modelers') "and/or the associated policy implications" ('policy wonks'). Was status as modelers vs. policy wonks predictive of performance, or is there too little data to say?

4) Figure 6 would benefit from an increased resolution if possible.

5) Despite my critique of asking about the "smallest, most likely, and largest possible number of cases" above, asking participants to consider the extremes of the distribution could slot into an elicitation protocol in other ways. Given that you found that experts gave too-narrow confidence intervals on their early esimates, in future studies you might consider a method similar to that described in Ch. 3 of "Developments in Demographic Forecasting" (eds. Mazzuco & Keilman), "Using Expert Elicitation to Build Long-Term Projection Assumptions" (Dion, Galbraith, and Sirag):

"(a) Experts are first asked to provide the lower and higher bounds of a range covering nearly all plausible values... Beginning with the contemplation of the extremes of the distribution is an intentional practice used to minimize potential overconfidence (Speirs-Bridge et al. 2010; Sperber et al. 2013; Oakley and O’Hagan 2014; Grigore et al. 2017; Hanea et al. 2018). Indeed, asking experts to first provide a single central estimate such as a mean or a median tends to trigger anchoring to that value in subsequent responses.

(b) Experts are asked to report how confident they are that the true value will fall

within the range they just specified in step 2(a). Allowing experts to determine

their own level of confidence has been found to reduce overconfidence in

comparison with asking them to identify the low and high bounds of an interval

to some predetermined confidence level (Speirs-Bridge et al. 2010).

(c) Experts are asked to estimate the median value of the plausible range they

provided in step 2(a), so that they expect an equal (50-50) chance that the true

value lies above or below the median.

(d) The range of values between the lower bound and the median is split in two

segments of equal length and the same is done for values between the median

and the upper bound. The respondent is then asked to assign to each segment

the probability that the true value falls within each of these segments. Note that

each half below and above the median has by definition 50% probability of

occurrence, so it is a matter of redistributing that 50% to each segment.

Throughout... several “checks”, in the form of pop-up warning signs, were

built in... in order to prevent illogical inputs in various forms."

This is not something that needs to be changed, just a thought to consider for future research.

Congratulations to the authors once again on an excellent piece of research.

Reviewer #2: review is uploaded as attachment

**Have the authors made all data and (if applicable) computational code underlying the findings in their manuscript fully available?**

Reviewer #1: Yes

Reviewer #2: **No: **missing regression data

PLOS authors have the option to publish the peer review history of their article (what does this mean?). If published, this will include your full peer review and any attached files.

Reviewer #1: **Yes: **Gabriel Recchia

Reviewer #2: No

Figure Files:

Data Requirements:

Reproducibility:

References:

---

## [Decision Letter · Decision Letter 1]

29 Mar 2022

Dear Dr. McAndrew,

Thank you very much for submitting your manuscript "An expert judgment model to predict early stages of the COVID-19 pandemic in the United States" for consideration at PLOS Computational Biology. As with all papers reviewed by the journal, your manuscript was reviewed by members of the editorial board and by several independent reviewers. The reviewers appreciated the attention to an important topic. Based on the reviews, we are likely to accept this manuscript for publication, providing that you modify the manuscript according to the review recommendations.

While Reviewer 1's comments were adequately addressed, Reviewer 2 has brought up numerous oversights and inconsistencies in the methodological descriptions in the SI that were not adequately addressed by the previous revisions or were newly introduced. While Reviewer 2 felt these concerns were serious enough to recommend rejection of the paper, the Editors believe the paper will be publishable after the authors address these remaining issues.  

Sincerely,

Alison L. Hill

Associate Editor

PLOS Computational Biology

Tom Britton

Deputy Editor

PLOS Computational Biology

[LINK]

Reviewer's Responses to Questions

**Comments to the Authors:**

Reviewer #1: Thank you! I'm satisfied with the way all of my comments were addressed. Congrats again on a nice piece of work.

Reviewer #2: Most of the cosmetic issues in the main text have been satisfactorily addressed. The SI is still unacceptable. The more I try to understand, the more confusing it becomes. “Consensus distribution” has been replaced by “linear pool” in the main text but not in the SI. There are 40 experts in tables S4 and S5, there are 41 experts on SI p. 7 and in the caption to Fig 5, there are 36 on github. Fig 5 caption says there are 40 “measurable questions “ in the right graph, where I count 44.

p.7 SI says “Experts who answered a survey for the first time or had no training data were assigned values equal to an unskilled forecaster , for example a relative entropy of one (i.e. the score of an unskilled forecaster). In later surveys, experts were not required to answer all questions, and in these cases, we assigned them the same value as an unskilled forecaster. These assignments enabled all responses to have observed data with which to calculate weights in a given week.” So some of the experts’ assessments are just invented? How many?”

SI tables 4 and 5 give 1207 as “Nr Observations”; presumably that means 1207 assessments of some question by some expert. Table 1 lists the number of experts answering each question. Summing the products (#experts * #questions) gives 1370.

The tables S4 and S5 are hard to interpret without some explanation of what the intercept and the beta’s represent. I don’t understand why only 4 of the expert betas are positive in each table. Half of the CI’s in S5 include zero (in S4, 13), meaning that the betas for “expert specific minus equal weights” are not significantly different from zero? Interpretation?

SI p.7: “Regression approaches in the past have had success making in sample and out of sample predictions on a diverse set of datasets [3, 4]” Neither of these references have anything to do with regression. Perhaps the authors confuse the linear pool with linear regression? In any case the differences here between performance weights and equal weights are extremely small. Since this is strongly at variance with a wealth of literature (google “performance weights expert judgment”) including recent pubs in PNAS, PLOS ONE (WHO) and Emerging Infection Diseases (CDC), the reader deserves some explanation. The factors mentioned on p 7 haven’t plagued other approaches. Perhaps the problem lies with the scoring variable. Indeed, rewarding honesty is not the same as rewarding goodness, as my little counter example pointed out. You don’t address counter examples by citing other people making the same mistakes.

**Have the authors made all data and (if applicable) computational code underlying the findings in their manuscript fully available?**

Reviewer #1: Yes

Reviewer #2: **No: **the data they provide is inconsistent

PLOS authors have the option to publish the peer review history of their article (what does this mean?). If published, this will include your full peer review and any attached files.

Reviewer #1: **Yes: **Gabriel Recchia

Reviewer #2: No

Figure Files:

Data Requirements:

Reproducibility:

References:

---

## [Decision Letter · Decision Letter 2]

28 Jul 2022

Dear Dr. McAndrew,

Thank you very much for submitting your manuscript "An expert judgment model to predict early stages of the COVID-19 pandemic in the United States" for consideration at PLOS Computational Biology.

As with all papers reviewed by the journal, your manuscript was reviewed by members of the editorial board and by several independent reviewers. In light of the reviews (below this email), we would like to invite the resubmission of a significantly-revised version that takes into account these comments.

We noticed that your Table S4 values have been changed in the most recent version of the manuscript. In your response in your revised paper please explain the reason for this. Also, the reviewer commented on some confusion with table and figure numbers; please check that these are correct.

We cannot make any decision about publication until we have seen the revised manuscript and your response to the reviewers' comments. Your revised manuscript is also likely to be sent to reviewers for further evaluation.

Sincerely,

Tom Britton

Deputy Editor

PLOS Computational Biology

Feilim Mac Gabhann

Editor-in-Chief

PLOS Computational Biology

Reviewer's Responses to Questions

**Comments to the Authors:**

Reviewer #1: I was asked to comment on the fact that "the authors have made unexplained changes to their tables which are pertinent to the critiques raised by the reviewer in the previous round ((in particular “SI_Redline.pdf”, Table 4, Table 5)". My comment is that the changes made seem to have been in response to the reviewer's comment that "The tables S4 and S5 are hard to interpret without some explanation of what the intercept and the beta’s represent", to which the authors responded that they had removed the intercept from tables S4 and S5. However on closer inspection it looks like the values on the initial lines (which had previously been described as "Intercept" but are now described as "Expert 0") are unchanged, while all other values have changed. I am guessing that the authors' code had previously erroneously used Expert 0 as the reference level, that they realized the error and updated the numbers with the correct analysis, but I might be wrong about this. I am happy to trust the authors' most recent analysis, although I would urge them to double-check that their latest numbers are indeed correct.

Something that may have added to the confusion is that there seem now to be two different Table S4s: one in Redline_Supporting_info.pdf ("Linear regression that compares the weights assigned to each expert using the expert-specific performance weighting and assigning experts equal weights"), and the "Table S4.pdf" uploaded separately - the latter seems to be the one referred to in the manuscript on p. 3 and p. 7. So I would also urge the authors to check over all their table, figure, and supplementary table/figure numbers. I don't think another round of review is required if changes of this sort are needed as this is the sort of thing that the authors can fix in proofing.

**Have the authors made all data and (if applicable) computational code underlying the findings in their manuscript fully available?**

Reviewer #1: Yes

PLOS authors have the option to publish the peer review history of their article (what does this mean?). If published, this will include your full peer review and any attached files.

Reviewer #1: No
---

## [Editor Report · Decision Letter 3]

11 Aug 2022

Dear Dr. McAndrew,

We are pleased to inform you that your manuscript 'An expert judgment model to predict early stages of the COVID-19 pandemic in the United States' has been provisionally accepted for publication in PLOS Computational Biology.

Best regards,

Tom Britton

Deputy Editor

PLOS Computational Biology

Tom Britton

Deputy Editor

PLOS Computational Biology

I am satisfied with the small modifications of the diagrams and support accepting the paper.

Kind regards, Tom Britton

---

## [Editor Report · Acceptance letter]

15 Sep 2022

PCOMPBIOL-D-21-01928R3 

An expert judgment model to predict early stages of the COVID-19 pandemic in the United States

Dear Dr McAndrew,

I am pleased to inform you that your manuscript has been formally accepted for publication in PLOS Computational Biology. Your manuscript is now with our production department and you will be notified of the publication date in due course.

With kind regards,

Zsofia Freund
